# Prognostic Significance of Lymph Node Ratio (LNR) in Gastric Cancer in Predicting Postoperative Complications and Survival: A Single-Center Study

**DOI:** 10.3390/cancers17050743

**Published:** 2025-02-22

**Authors:** Michał Miciak, Krzysztof Jurkiewicz, Przemysław Dzierżek, Julia Rudno-Rudzińska, Wojciech Kielan

**Affiliations:** Clinical Department of Oncological Surgery, University Centre of General and Oncological Surgery, Faculty of Medicine, Wroclaw Medical University, 50-556 Wrocław, Poland; krzysztof.jurkiewicz@student.umw.edu.pl (K.J.); julia.rudno-rudzinska@umw.edu.pl (J.R.-R.); wojciech.kielan@umw.edu.pl (W.K.)

**Keywords:** lymph node ratio, gastric cancer, gastrectomy, prognosis, overall survival, complications after gastrectomy

## Abstract

The Lymph Node Ratio (LNR) index appears as a promising parameter for predicting prognosis in gastric cancer. This parameter represents the proportion of metastatic to all examined lymph nodes. It could potentially improve the accuracy of survival determination more specifically than the N feature of TNM classification. Our retrospective study analyzed the correlations between obtained LNR, gastric cancer characteristics, surgical procedure, postoperative complications, and survival rate. The studied parameter was strongly correlated with gastric cancer severity and negatively with patient survival. Our findings could have relevant clinical implications in gastric cancer management: LNR, if validated, could enhance future prognostic algorithms, leading to more informed treatment decisions and patient outcomes.

## 1. Introduction

Gastrectomy is strongly associated with Jan Mikulicz-Radecki, the patron of the University Clinical Hospital in Wroclaw. This Polish–German surgeon, who lived in the late 19th and early 20th centuries, developed strict guidelines for performing this surgical procedure and significantly increased the diagnosis of gastric cancer in the early stages. The gastrectomy procedures he performed at the time resulted in a permanent cure rate of more than 60%. By a perverse fate, the cause of his death was inoperable gastric cancer [1]. It should be unequivocally emphasized that gastric cancer is currently a significant surgical and oncological problem. Worldwide, it is the fourth most common cancer and the second most common cause of cancer deaths [2,3]. There is currently a decline in mortality and morbidity, but projections indicate a future long-term increase in the incidence of the disease. The incidence in Eastern Europe currently stands at 0.025% of the population [4,5]. Gastric cancer accounts for about 4% of all cancers detected. In 2019 in Poland, gastric cancer was the seventh most common cancer in men and the thirteenth in women. It accounted for 3.8% and 2.2% of all cancers, respectively. Statistically, more than 90% of patients were over the age of 50. In both men and women, the most common age for gastric cancer incidence was 65–69 years (664 and 292 cases, respectively). Gastric cancer was also the fifth most common fatal cancer in the male group and the seventh in the female group. Analyzing these data, it can be concluded that it is absolutely necessary to effectively diagnose and treat gastric cancer patients [6,7,8]. The basis of gastric cancer diagnosis remains as endoscopic examination, and the current treatment preference is neoadjuvant chemotherapy lasting up to 3 months (mainly in the FLOT regimen), followed by surgery. The type of procedure depends mainly on the location of the tumor and the depth of invasion. In general, it includes endoscopic mucosal resection, distal esophagectomy, and subtotal or total gastrectomy with potential adjuvant chemotherapy administration [9]. Before treatment, every patient must undergo imaging tests, including a CT scan of the chest, abdomen, and pelvis with intravenous contrast, as well as an endoscopic examination of the upper gastrointestinal tract with biopsy sampling for histopathological and microbiological analyses. Imaging studies are essential for assessing the tumor infiltration of adjacent organs and detecting distant metastases. In the case of unresectable or disseminated tumors, each biopsy sample is additionally tested for HER2 receptor status and microsatellite stability. These preoperative procedures are essential for determining the extent of the tumor as well as accurate staging, and selecting the most appropriate treatment approach. For patients eligible for radical treatment, the recommended approach includes the aforementioned neoadjuvant chemotherapy in the FLOT regimen, followed by surgery with D2 lymphadenectomy. D1/D1+ lymphadenectomy is acceptable only for low-grade early T1a and T1b tumors (unless the diameter of the lesion exceeds 1.5 cm) with no lymph node metastases (N0). In cases where palliative treatment is required, systemic therapy is tailored accordingly (e.g., trastuzumab in combination with a platinum derivative and fluoropyrimidine for HER2-positive tumors), along with either palliative resection or bypass procedures. For palliative resection, it is not recommended to perform lymphadenectomy outside the D1 range [10]. Patients frequently do not present with specific symptoms in the early stages of gastric cancer. However, as the disease progresses, symptoms become more pronounced and concerning. This highlights the importance of timely diagnosis, as delays can lead to poorer patient prognosis [11]. A commonly used method for assessing the progression of cancer is the TNM scale, which includes the clinical staging of the disease based on the advancement of its features. According to the eighth edition of the TNM classification designed by the UICC (Union for International Cancer Control) and AJCC (American Joint Committee on Cancer), to adequately assess lymph node metastasis (N feature), it is recommended to examine at least 16 lymph nodes, and examining at least 30 nodes significantly improves the accurate determination of prognosis [12,13]. In early gastric cancer with invasion, a threshold of >25 removed lymph nodes is recommended for a correct diagnosis [14]. Finally, the Lymph Node Ratio (LNR) is the proportion of lymph nodes affected to lymph nodes removed [15]. It appears to be a more accurate indicator than only the N feature of the TNM classification, which can sometimes overestimate an unfavorable prognosis. In addition, LNR does not necessarily require the removal of more than 15 lymph nodes, which seems to be necessary in classic TNM staging for the prognosis to be reliable [16]. Thus, LNR is a reliable parameter for predicting the survival of patients with gastric cancer, as demonstrated in a meta-analysis by Zhu J. et al. However, it has not yet been included in management guidelines, though its addition has been proposed [17]. The purpose of this study is to investigate and confirm LNR utility in estimating prognosis, tumor advancement, and survival, as well as its association with Clavien–Dindo (C-D) grading of postoperative complications in patients treated for gastric cancer, based on the experience of a single European center.

## 2. Materials and Methods

The study included a retrospective analysis of 194 patients operated on for gastric cancer at the Clinical Department of Oncological Surgery, University Centre of General and Oncological Surgery of the University Clinical Hospital in Wroclaw between January 2017 and December 2021 (Figure 1).

Patients were evaluated in terms of age, sex, tumor location, TNM classification, tumor stage, surgical procedure, chemotherapy administration, number of lymph nodes removed, C-D grading, survival time, and LNR. All patients underwent blood tests, thoracic X-ray or CT scan, abdominal and pelvic CT scan, and gastroscopy with microbiological and histopathological sampling, with the identification of gastric tumor location and staging. The chemotherapy regimens included FLOT (docetaxel + oxaliplatin + leucovorin + 5-fluorouracil), EOX (epirubicin + oxaliplatin + capecitabine), and ECX (epirubicin + cisplatin + capecitabine), with FLOT being the regimen administered most commonly. D2 lymphadenectomy was performed during the gastrectomy procedures included in our study. A more radical approach, such as posterior and para-aortic D2+ lymphadenectomy, may provide better loco-regional control in advanced gastric cancer stages with a high risk of metastasis and offer satisfactory risk stratification for perioperative procedures [18]. However, in our center, the standard approach was D2 lymphadenectomy, performed in accordance with the current guidelines [10]. Removed lymph nodes were histopathologically examined, counted, and numbered according to the Japanese Research Society for Gastric Cancer classification (Table 1) [19]. Postoperative complications were assessed using the C-D grading system: grade I represents minor deviations from the standard postoperative course (administration of antiemetics, analgesics, diuretics, electrolytes, or physiotherapy), grade II represents complications requiring pharmacotherapy other than previously mentioned (including blood derivatives or parenteral nutrition), grade III represents complications requiring surgical, endoscopic, or radiological intervention (with or without general anesthesia), grade IV represents directly life-threatening complications (multiple organ failure, renal failure with hemodialysis requirement, central nervous system complications), and grade V represents patient death. All C-D grades refer to complications occurring up to 30 days after the surgical procedure [20].

### Statistical Analysis

Statistical analysis was performed using IBM SPSS Statistics 28. The association of LNR with patient demographics, gastric cancer characteristics, C-D grade, and survival was verified using Student’s t-tests for independent samples and non-parametric Mann–Whitney U and Kruskal–Wallis H tests for quantitative variables due to the unequal distribution of the data compared. In addition, Pearson’s r correlation analyses and chi-square tests of independent comparisons were performed for nominal variables. A significance threshold of *p* < 0.05 was applied where appropriate.

## 3. Results

Among the 194 patients, there were 136 men (70.1%) and 58 women (29.9%). The average age of the entire population was 67 years, which was equal for men and women. The youngest patient was 28 years old, the oldest 93 years old. In total, 100 patients (51.5%) underwent a cycle of perioperative chemotherapy and 94 patients (48.5%) did not undergo chemotherapy. Of the patients, 81 (41.8%) presented with a proximally located lesion (tumor of the cardiac orifice, fundus, or upper part of the gastric body), and 63 patients (32.5%) presented with a distally located lesion (tumor of the pylorus, pyloric outlet, or lower part of the gastric body). In 50 (25.7%) patients, the location of the tumor was considered undetermined. Total gastrectomy was performed in 77 patients (39.7%), including 59 men and 18 women. Subtotal gastrectomy was performed in 56 patients (28.9%), including 36 men and 20 women. Resection was abandoned in 61 patients (31.4%), including 41 men and 20 women. For TNM classification, subgroups a and b for T1, T4, and N3 features were abandoned due to insufficient study groups. The highest number of patients (32; 16.5%) were diagnosed with TNM stage IA, while the lowest number (8; 4.1%) presented TNM stage 0 or IIIA. Upon post-operative evaluation, most patients (81; 41.8%) presented with C-D grade I, and C-D grade IV occurred with the lowest frequency (4; 2.1%). All demographic and clinical patient data are summarized in Table 2.

Finally, 133 patients with resectable lesions proceeded to the further part of the study and the main statistical analysis. Based on the histopathological results of patients undergoing gastrectomy procedures with D2 lymphadenectomy, the number of lymph nodes removed and affected were summarized and LNR values were calculated. During gastrectomy procedures, 3347 lymph nodes were removed, with 440 (13.1%) lymph nodes affected (general LNR = 0.131). The average patient LNR was 0.134; 36 patients had ≤15 nodes removed and 97 had ≥16. In terms of total quantities, 11.8% and 16.3% of the removed nodes were affected, respectively. The largest number of affected nodes belonged to station No. 3 (lesser curvature nodes), with 134. This was 30% of the affected and 3.6% of all removed nodes. The second most affected station was No. 4 (greater curvature nodes), with 113, and the third was No. 6 (infrapyloric nodes) with 29. No affected nodes were found at station No. 14 (superior mesenteric vein/artery nodes). Details are presented in the aforementioned Table 1. The largest number of nodes removed was in a 51-year-old patient (total gastrectomy) (67). Seventy patients (53%) had no affected nodes after gastrectomy. The largest number of affected nodes (41/41; 100%) was found in a 57-year-old patient after total gastrectomy. On average, 25 nodes were removed and 3 nodes were affected.

The results of the analyses of LNR variation for TNM features, lesion location, and C-D scale are presented in Table 3. C-D grades III, IV, and V were aggregated into a combined group for a more reliable analysis due to the small number of study samples. The analyses confirmed the presence of statistically significant differences for TNM features. There was a significantly higher LNR in the T4 group compared to T0 and T1 (*p* < 0.05). The LNR was also lower for T1 compared to T3. Due to the small number of observations, Tis had no significant differences compared to the other groups, and similarly, T2 did not differ from the other groups. A higher level of the T feature is associated with a higher LNR. In terms of the N feature, significant differences were also confirmed, which indicated a significantly lower LNR for N0 compared to the higher stages (*p* < 0.05). N1 had a lower LNR than N3 (*p* < 0.05). In contrast, there were no differences between N2 and N1 (*p* = 0.387) or N3 (*p* = 0.983). A higher LNR for M1 compared to M0 was also confirmed (*p* < 0.05). Note that here, the M1 stage of a resectable lesion indicates the presence of distant metastases in other stations (station Nos. 1–12 and 14v are regional lymph nodes). The analyses did not confirm statistically significant differences between lesion location and C-D grade in relation to LNR (*p* = 0.974). Regardless of the location of the lesion and C-D grade, the LNR tended to range from 0 to 0.20, with an average of about 0.15.

Pearson’s r correlation analyses of the association between LNR and cancer stage, age, and patient survival in months are presented in Table 4. LNR was not dependent on age, while significant associations were found between cancer stage and survival (*p* < 0.05). Higher stage was associated with higher LNR. In addition, patients exhibited lower survival as LNR increased. Thus, lower survival was also associated with gastric cancer stage.

Figure 2 presents the LNR variation analysis using Student’s t-test for variables that complied with the assumptions of sampling equality and equality of variance. There were no statistically significant differences in LNR in relation to procedure type (*p* = 0.714), sex (*p* = 0.929), or perioperative chemotherapy (*p* = 0.356). This means that regardless of these factors, the average LNR ranged from 0.12 to 0.15 in each group.

As there was no significant relationship between LNR and C-D grade, an analysis of the association of C-D complication grade with survival expressed in months was performed (Table 5). It was confirmed that 30% more patients survive if they present with C-D I or II compared to C-D III or higher, and the average survival between the study groups was one year more for C-D I and II than C-D III and higher (*p* < 0.05). In contrast, there were no significant differences between the length of survival for C-D I and II (*p* = 0.109) or C-D II and III or higher (*p* = 0.789).

## 4. Discussion

Gastric cancer, as an important problem in oncological surgery, requires comprehensive diagnosis and treatment. Analysis of the data on this malignancy shows that it is the fourth leading cause of cancer death worldwide, and its 5-year survival rate oscillates between 38 and 43% with an upward trend [21]. This is probably related to the increasingly effective diagnosis of gastric cancer at an early stage, resulting in therapeutic success. Many prognostic factors are involved in predicting the prognosis after gastrectomy, e.g., patient age, type of surgical procedure, TNM tumor stage, C-D grade of postoperative complications, and LNR. TNM scale and LNR appear to be the basis of tumor evaluation, but successive studies have shown specific limitations [12,13,14,15,16]. The number of lymph nodes retrieved is an important parameter, on the basis of which the appropriate index is chosen. As mentioned, LNR appears to be statistically significant in predicting prognosis, even when the number of nodes examined does not exceed 15, whereas this value is described as the low-optimal value for correct TNM determination [16].

The present study included patients diagnosed with gastric cancer that qualified for surgical treatment via total or subtotal gastrectomy with D2 lymphadenectomy. In the group of patients with unresectable lesions, LNR could not be used due to the inability to retrieve nodes for examination. In the analysis of the relationship between LNR and TNM features, we noted that the index increases with higher TNM stages. LNR was significantly higher for the T4 stage, compared to T1 and T0 (*p* < 0.05). Díaz del Arco C. et al. also observed increases in LNR with higher tumor stage. The authors also mention the upward dependence of LNR on lymphovascular invasion, perineural infiltration, and infiltrative tumor growth, with these parameters being associated with tumor size and, consequently, higher T stages [22]. Fukuda N. et al. concluded that higher LNR values (according to the cutoff thresholds used by the authors) are associated with higher pT stages and larger tumor diameters, and these characteristics significantly affect 5-year survival rate [23]. For the N feature, in our analysis, LNR assumed higher values for N3 compared to N1 and N0 (*p* < 0.05). This relationship appears to be statistically significant based on the definition of LNR and N staging in gastric cancer. This is related to the present metastasis of advanced tumors (T4, N3), which is reflected in the number of affected lymph nodes and higher LNRs. Giuffrida M. et al. studied the association between LNR and the quantity of metastatic nodes alone and came to similar conclusions: LNR was higher in patients with a higher N stage. In contrast, this study also found that survival rates were not associated with increasing N stage, while they were with higher LNRs. The analysis proposed LNR as an independent prognostic indicator (*p* < 0.05), but noted the necessity of improvements in this area, like the determination of universal LNR intervals [24]. Certain studies propose specific cutoff values and prognostic models for patients with gastric cancer, for example, a min. of 0.80 with a min. of 15 nodes with present metastases, which is supposed to be associated with a significant decrease in overall survival [25]. In patients in the M1 stage, LNR was also found to be statistically significant compared to the M0 stage (*p* < 0.05). In the literature, there are limited similar examples of the higher survival of patients with distant metastatic gastric cancer who presented lower LNR values, and thus limited examples of the resulting superiority of LNR over only the number of nodes affected. Further novel indicators, like log odds of positive lymph nodes (LODDS), are also being analyzed [26]. The M1 stage is usually considered as disseminated and inoperable cancer. As mentioned before, in our study, patients who underwent resection and were classified as M1 presented metastases outside the regional lymph nodes (station Nos. 1–12 and 14v) [27]. Thus, the lesion should be classified in the M1 stage when the nodes of these stations are affected, but the LNR remains unchanged. We observed distant metastasis to the lymph nodes at station Nos. 13, 15, and 16 (retropancreatic, middle colic, and para-aortic nodes, respectively). Out of 440 total affected nodes, 11 lymph nodes at these stations were affected; therefore, this was not an amount that could significantly impact the studied LNR correlations.

When analyzing LNR according to age, sex, surgical procedure, chemotherapy administration, lesion location, and C-D grade, we found that LNR was not statistically significant in relation to any of these variables (*p* = 0.481; 0.929; 0.714; 0.356; 0.924; and 0.974, respectively), and ranged from 0.12 to 0.15 among these factors. Many studies also do not describe direct relevant LNR relationships with the aforementioned parameters. Despite this, LNR is linked to tumor progression, so it is indirectly related to increasing TNM stages and advanced age of the patient, when lesions are frequently already disseminated. Despite the limited information about LNR’s direct correlation with C-D grade reported in the literature, a more advanced tumor stage would be characterized by a separate increase in both parameters. The requirement for a widespread lymphadenectomy may be associated with a longer operative time, expansion of the operative field, and consequently a higher C-D grade, along with a more accurately obtained LNR. Thus, LNR has the potential to indirectly predict the risk of postoperative complications. Considering the balance of benefits associated with extended lymphadenectomy, we recommend performing D2 lymphadenectomy and predicting complications based on the obtained LNR, rather than performing extended procedures and risking more perioperative complications. Ultimately, all the presented parameters are described as prognostic factors for gastric cancer regardless of LNR inclusion [28,29,30,31,32]. Our analysis also revealed that 30% more patients survive if they exhibit Clavien–Dindo grades I and II, compared to grade III or higher (*p* < 0.05). C-D grade II is already thought to significantly worsen the prognosis [33]. Another analyzed parameter was the clinical cancer stage, which follows directly from the TNM classification. Its correlation with LNR (confirmed by Pearson’s r coefficient) was also statistically significant (*p* < 0.05), which validates the correlation of higher LNR with higher tumor stages like IIIA, IIIB, IIIC, or IV. LNR is also dependent on the stage of gastric cancer in other classifications, such as Borrmann’s or Lauren’s, and takes on higher values with increasing cancer malignancy in these classifications [34,35]. Cancer stage has also been proven to be inversely related to overall survival. This relationship is noted by numerous authors [12,28,36]. As noted, the LNR value was not dependent on the administration of perioperative chemotherapy in our study. This might result from the regression of metastatic nodes after neoadjuvant treatment. Moreover, in cases with less than 15 nodes retrieved, TNM stage alone may reveal a correlation between these factors and expose errors in the prediction of prognosis [36]. Down-staging of the disease is also possible with the use of highly effective chemotherapy, such as pressurized intraperitoneal aerosolized chemotherapy (PIPAC), in the case of disseminated gastric cancers. This method achieves a high concentration in peritoneal deposits and is effective in the treatment of peritoneal metastasis [37]. Establishing the LNR value is now being emphasized to inform the decision to administer chemotherapy to prevent tumor recurrence [38]. However, there are confirmed correlations of the relationship between LNR and chemotherapy in the literature. Sakin A. et al. proved that LNR was associated with the length of overall survival in patients treated with neoadjuvant chemotherapy, depending on the cutoff point established in the analysis, with a value of 0.255. The study by Zhou P. et al. focused on neoadjuvant immunochemotherapy, where overall survival was also higher in patients with lower LNR; in this case, the cutoff point was 0.330. Both authors agree that LNR is a more reliable indicator in patients treated with chemotherapy than the TNM number of involved nodes alone [39,40]. Finally, as LNR increased, patient survival (expressed in months) decreased, and this relationship was confirmed by a negative Pearson’s r coefficient (*p* < 0.05). The mean LNR in our study was 0.134; patients with equal or lower values survived an average of 32.5 months after surgery, while patients with a higher value survived an average of 19.9 months. This gives a difference of 12.6 months in mean survival in favor of patients with a lower LNR. There are no set rigid cutoff points for LNR considered significant in the literature. Various authors set different cutoff points, classifying patients into low-LNR and high-LNR groups, and the value varies depending on the study. However, a common finding is the lower survival of patients with an established high LNR, which confirms the results of our study. LNR is considered as an independent prognostic marker, and is significantly associated with poorer survival [12,36,39,40,41]. A large multicenter study by Marchet A. et al. reported that the best cutoffs for LNR groups could be as follows: 0.00; 0.00–0.01; 0.01–0.09; 0.1–0.25; and >0.25 LNR rate. The proposed or similar divisions have also been utilized by certain other authors, and the results are described as statistically significant [30,42,43]. Higher cutoff points for LNR may be associated with up to several times increased mortality. In the study by Topcu R. et al., the study group with an LNR of 0.32 or higher had a 4.8 times higher mortality rate than the study group with an LNR lower than the established value. The values expressed in months were 24.22 and 48.01 months, respectively. The results were statistically significant and further confirm the importance of determining LNR [44]. As mentioned at the beginning of this study, LNR independence as a prognostic factor in opposition to the TNM classification and the possibility of including this parameter in future tumor staging classification systems are currently emphasized. New studies using statistical models aim to find an appropriate common accuracy for LNR and TNM classification [17,45].

### Limitations and Future Directions

This study was a retrospective analysis; therefore, the drawbacks of this type of research could not be avoided. For example, the process of patient inclusion is not as accurate as in prospective studies. In this study, we did not take into consideration, e.g., the technique of the procedure (laparoscopic/open), which also has an impact on length of hospitalization, postoperative inflammatory response, and resulting complications [46]. Furthermore, this was a single-center study, so the number of patients is not as representative as in multicenter studies. Future research directions would include studies on larger groups of patients with established low-LNR and high-LNR values, and analyzing the results with, e.g., molecular tests or other promising indicators for gastric cancer that are being studied. The neutrophil–lymphocyte ratio (NLR) or platelet–lymphocyte ratio (PLR) may become useful in selecting appropriate treatment options alongside LNR, which is a valuable future research direction [47].

## 5. Conclusions

LNR is related to TNM classification features and gastric cancer clinical stage, and correlates negatively with patient survival. The parameter increases for advanced gastric lesions (T3–T4, N3, M1), while it remains relatively constant for lower-grade lesions. LNR is a more sensitive prognostic indicator than the quantity of nodes involved alone, as it does not always require a minimum threshold of examined nodes. A higher LNR is associated with a poorer prognosis, but there are no definitively established cutoff points for LNR values in gastric cancer. In the absence of LNR, C-D complication grade could be a prognostic parameter, with grade III or higher being associated with a poorer prognosis. The independence of LNR as a prognostic indicator has been given significant attention in studies, establishing the possibility of implementing LNR in future tumor staging classification systems.

## Figures and Tables

**Figure 1 cancers-17-00743-f001:**
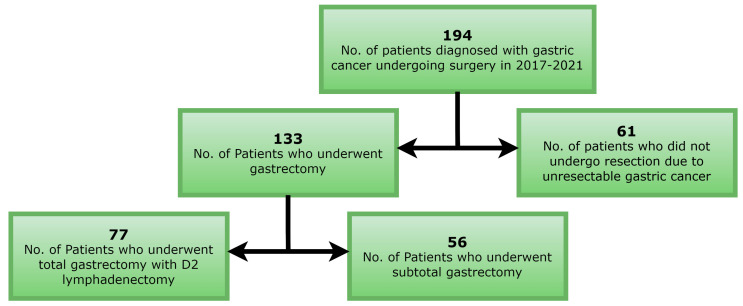
Study group selection process.

**Figure 2 cancers-17-00743-f002:**
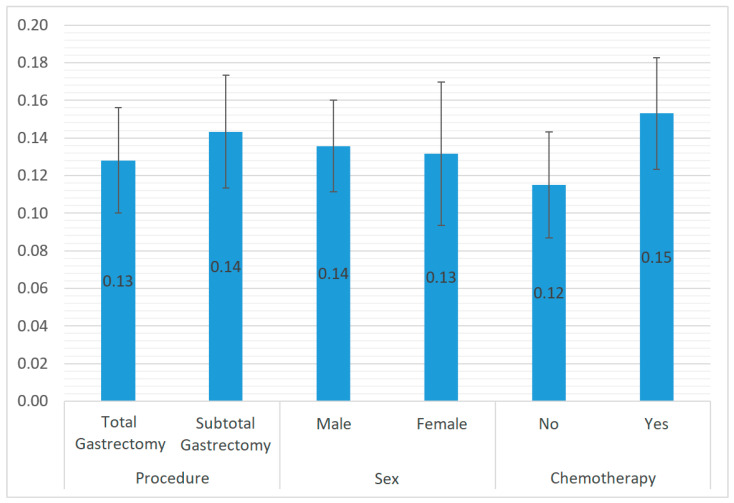
LNR variations depending on procedure type, sex, and chemotherapy.

**Table 1 cancers-17-00743-t001:** Numbering of lymph node stations according to the classification of the Japanese Research Society for Gastric Cancer, along with the quantity of affected lymph nodes. LNs—lymph nodes.

Station No.	Station Name	Quantity of Affected LNs
1	Right cardiac LNs	22
2	Left cardiac LNs	16
3	LNs along the lesser curvature	134
4	LNs along the greater curvature	113
5	Suprapyloric LNs	18
6	Infrapyloric LNs	29
7	LNs along the left gastric artery	19
8	LNs along the common hepatic artery	19
9	LNs around the celiac artery	23
10	LNs at the splenic hilus	11
11	LNs along the splenic artery	9
12	LNs in the hepatoduodenal ligament	16
13	LNs at the posterior surface of the pancreatic head	2
14v/14a	LNs along the superior mesenteric vein/artery	0
15	LNs along the vessels of the middle colon	7
16	Para-aortic LNs	2

**Table 2 cancers-17-00743-t002:** Demographic and clinical characteristics of patients.

Feature		No. of Patients
Sex	Male	135 (69.6%)
	Female	58 (30.4%)
Age (years)	≤67	90 (46.4%)
	>67	104 (53.6%)
Tumor Location	Proximally	81 (41.8%)
	Distally	63 (32.5%)
	Not Specified	50 (25.7%)
Type of surgery	Total Gastrectomy	77 (39.7%)
	Subtotal Gastrectomy	56 (28.9%)
	Non-Resectable	61 (31.4%)
CTH	Yes	100 (51.5%)
	No	94 (48.5%)
TNM stage	0	8 (4.1%)
	IA	32 (16.5%)
	IB	12 (6.2%)
	IIA	18 (9.3%)
	IIB	19 (9,8%)
	IIIA	8 (4.1%)
	IIIB	14 (7.2%)
	IIIC	9 (4.6%)
	IV	13 (6.7%)
	Not Specified	61 (31.5%)
T	T0	6 (3.1%)
	Tis	2 (1.0%)
	T1	39 (20.1%)
	T2	17 (8.7%)
	T3	38 (19.6%)
	T4	31 (16.0%)
	Not Specified	61 (31.5%)
N	N0	70 (36.4%)
	N1	22 (11.3%)
	N2	17 (8.9%)
	N3	23 (11.9%)
	Not Specified	61 (31.5%)
M	M0	120 (61.9%)
	M1	13 (6.6%)
	Not Specified	61 (31.5%)
C-D Grade	I	81 (41.8%)
	II	72 (37.1%)
	III	16 (8.2%)
	IV	4 (2.1%)
	V	21 (10.8%)

Annotation. CTH—chemotherapy; C-D—Clavien-Dindo.

**Table 3 cancers-17-00743-t003:** Variation in LNR according to TNM features, lesion location, and C-D complication grade.

Feature	M	SD	Q1	Me	Q3	Test Value	df	*p* Value
T feature								
T0	0.00	0.00	0.00	0.00	0.00	35.14	5	<0.05
T1	0.03	0.09	0.00	0.00	0.00			
T2	0.09	0.17	0.00	0.00	0.11			
T3	0.18	0.28	0.00	0.06	0.22			
T4	0.27	0.29	0.00	0.21	0.47			
Tis	0.00	0.00	0.00	0.00	0.00			
N feature								
N0	0.00	0.00	0.00	0.00	0.00	127.47	3	<0.05
N1	0.06	0.04	0.03	0.04	0.09			
N2	0.23	0.18	0.12	0.20	0.22			
N3	0.54	0.26	0.32	0.50	0.80			
M feature								
M0	0.11	0.21	0.00	0.00	0.12	1104.50	1	<0.05
M1	0.36	0.37	0.00	0.22	0.65			
Lesion location								
Distally	0.14	0.22	0.00	0.00	0.21	2194.50	1	0.924
Proximally	0.13	0.25	0.00	0.00	0,13			
C-D complications								
Grade I	0.11	0.20	0.00	0.00	0.13	0.05	2	0.974
Grade II	0.16	0.27	0.00	0.00	0.21			
Grade III or higher	0.16	0.28	0.00	0.00	0.16			

Annotation. The test values refer to the Kruskal–Wallis H-test when comparing more than two groups, while comparisons for two groups use the Mann–Whitney U-test value. Q1—first quartile; Me—median; Q3—third quartile; C-D—Clavien–Dindo.

**Table 4 cancers-17-00743-t004:** Analysis of the relations between LNR and age, cancer stage, and patient survival time.

Feature		Age (Years)	Cancer Stage	Survival (Months)
LNR	Pearson’s r	0.06	0.65	−0.36
	*p* Value	0.481	<0.05	<0.05

Annotation. LNR—lymph node ratio.

**Table 5 cancers-17-00743-t005:** Relationship between C-D complication grade and survival time in months.

Feature	M	SD	Q1	Me	Q3	Test Value	df	*p* Value
C-D Grade I	33.80	17.04	21.00	34.00	48.00	11.52	2	<0.05
C-D Grade II	26.09	18.66	11.00	22.00	36.00			
C-D Grade III or higher	21.11	20.25	1.00	16.00	41.00			

Annotation. The test values refer to the Kruskal–Wallis H-test when comparing more than two groups, while comparisons for two groups use the Mann–Whitney U-test value. Q1—first quartile; Me—median; Q3—third quartile; C-D—Clavien-Dindo.

## Data Availability

The datasets used and/or analyzed during this study are available from the corresponding author upon reasonable request, and are not publicly available due to privacy and ethical restrictions.

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
