# Peer review of "Prognostic Significance of Lymph Node Ratio (LNR) in Gastric Cancer in Predicting Postoperative Complications and Survival: A Single-Center Study"

_cancers, 2025, doi:10.3390/cancers17050743_

Round 1
Reviewer 1 Report
Comments and Suggestions for Authors
Our surgical colleagues propose their paper whose aim is to evaluate not the total number of positive lymph nodes examined histologically, but the percentage of positive lymph nodes on the total number. Excellent absract that represents an excellent summary of the work. The introduction begins with a dutiful and rightly respectful tribute to Prof. Jan Mikulicz-Radecki, who spent his entire life studying stomach cancer. Also in the introduction it is recommended to write something about the diagnostic pathway of patients who were diagnosed with gastric cancer with biopsy and whether microbiological research was done to immediately know the stability or otherwise of microsatellites and HER 2. This information is essential for cutting-edge neoadjuvant treatment. Then important is the imaging study to understand what radiological stage the patient is in. Only after this information can we possibly start the patient on a resective surgical pathway. The materials and methods are well described, in particular D2 resection is discussed and then the removal of the para-aortic lymph nodes is described, we would like to know if the posterior and intercavo-aortic D2 plus nodes were removed, where it was indicated, obviously, (doi.org/10.3390/cancers16071376 to be read and cited in the bibliography). Excellent discussion of which we disagree only on one point, when there are secondary lesions especially peritoneal one can think of a down-staging with the administration of chemotherapeutic drugs with PIPAC whose effects are present in the bibliography (doi: 10.1016/j.critrevonc.2022.103846 to be read and cited in the bibliography). For the rest good iconography, good English, good bibliography.
Reviewer 2 Report
Comments and Suggestions for Authors
This is study of 194 patients with gastric cancer to determine the use of the lymph node ratio (LNR) with positive findings in predicting prognosis and survival time. This study adds to the validation of the LNR in care of patients with gastric cancer.
The abstract is complete and provides a comprehensive review of the study and results.
In the introduction/background, there is information on the guidelines for the gastrectomy procedure, but it doesn’t seem to directly relate to this study. There is good information related to the prevalence of gastric cancer. Treatments, chemotherapy and surgery, are noted but do not provide information directly related to the study. While there is some discussion of the use of the LNR, the references are over ten years old while a number of newer studies have been conducted but are not mentioned until the discussion. This leads one to believe this is still a novel approach while further validation is needed, but there are several articles, including the meta-analysis noted in the discussion.
The methods are described well and the consort diagram is helpful. The sampling is clear. The retrospective data collected were clearly described. The general statistical analysis are correct.
The results of the demographics of the patients are clearly described and presented in Table 2. There is quite a bit of the table presented in the text. This seems redundant. One might highlight some of the main findings.
There is presentation of the overall LNR for those with total and subtotal gastrectomy, but it isn’t clear what the meaning of this aggregating of data represents given the aim of the study. There is good description of the number, location, and status of the lymph nodes and shown in Table 3. There is detailed presentation of the comparison of the LNR to TNM feature based on lesion location and C-D complication grade.
In the discussion, it is noted that the TNM and LNR scales have specific limitations based on studies, but no references are provided. In the discussion of the use of the LNR several newer studies have not been included such as those by Ergenc et al. (2023), Alakus et al. (2021), Topcu et al. (2022).
There seems to be some assumptions related to the relationship between widespread lymphadenectomy and a higher C-D grade on page 8, lines 263-269. Is this important to the study results, especially in light of the results?
Limitations are appropriately stated.
The conclusions relate directly to the results.
Reviewer 3 Report
Comments and Suggestions for Authors
The authors analyze a topic which is of interest – Prognostic significance of Lymph Node Ratio (LNR) in Gastric Cancer in predicting postoperative complications and survival. It has been analyzed for more than a decade, but mainly in Asia so a single center study from central Europe has its interest.
The presentation is clear, comprehensive and well documented.
Some observations:
- In the introduction in row 65 the authors write:
A number of patients frequently do not present with specific symptoms in the early stages of gastric cancer, which can lead to a poorer prognosis due to insufficiently radical resection [10].
citing
Wang, H.; Qi, H.; Liu, X.; Gao, Z.; Hidasa, I.; Aikebaier, A.; Li, K. Positive lymph node ratio is an index in predicting prognosis 380 for remnant gastric cancer with insufficient retrieved lymph node in R0 resection. Sci Rep. 2021, 11: 2022. doi:10.1038/s41598- 381 021-81663-0.
Who wrote:
Furthermore, most RGC patients are in the advanced stage because of the lack of specific symptoms during the early stage, which leads to a lower radical resection rate and poor prognosis5 Citing:
5.Zhang Y, et al. Surgical treatment of gastric remnant-stump cancer. J. Nippon. Med. Sch. 2002;69(5):489–493. doi: 10.1272/jnms.69.489
So, in my opinion , the statement : A number of patients frequently do not present with specific symptoms in the early stages of gastric cancer, which can lead to a poorer prognosis due to insufficiently radical resection [10], is with no interest for the present paper as the cited papers deal with remnant gastric stump cancer.
- The authors write:
During 140 gastrectomy procedures, 3347 lymph nodes were removed, with 440 (13.1%) lymph nodes 141 affected (LNR = 0.131). In the total gastrectomy group, 2364 lymph nodes were removed, 142 with 279 lymph nodes affected (LNR = 0.118). In the subtotal gastrectomy group, 983 143 lymph nodes were removed, with 161 lymph nodes affected (LNR = 0.164).
I see no interest in giving such figures with total lymph nodes harvested (3347) in 140 procedures, but maybe they have a logic explanation.
- In row 153: 70 patients after gastrectomy (53%) had none affected nodes
According to the Modern Language Association (MLA), one should never begin a sentence with a numeral. Instead, one should try to reword the sentence or spell out the number.
- The authors write:
In this study we retrospectively analyzed 194 patients diagnosed with gastric cancer operated on between 2017 and 2021 but the Survival data used for the publication are contained in the timeframe up to 29.08.2022. So the last patients had a follow up less than a year – maybe a longer follow-up would have been suitable as one of the main goals of the study is survival.
The 2 figures are mandatory for the study design and for presenting the LNR variations depending on procedure type, sex and chemotherapy.
The 6 tables offer concentrated information on the results, from all the necessary points of view. Table 2 is on 2 pages and a little bit too wide. Table 3 and 4 are too wide.
I found no plagiarism.
The discussions and conclusions are sincere, coherent and connected to the content.
The references are appropriate, up-to-date and contain 41 titles.
I found no self-citations.
